Face matching and metacognition: investigating individual differences and a training intervention

http://orcid.org/0000-0001-8339-8832 Kramer Robin S. S. remarknibor@gmail.com
School of Psychology, University of Lincoln , Lincoln, Lincolnshire , United Kingdom
Barnhart Anthony
Electronic publication date: 2023 Jan 25
Publication date: 2023
Volume: 11
Electronic Location ID: e14821
Received 2022 Sep 9; Accepted 2023 Jan 6
Copyright: © 2023 Kramer
Copyright year: 2023
Copyright holder: Kramer
License: This is an open access article distributed under the terms of the Creative Commons Attribution License, which permits unrestricted use, distribution, reproduction and adaptation in any medium and for any purpose provided that it is properly attributed. For attribution, the original author(s), title, publication source (PeerJ) and either DOI or URL of the article must be cited.
License URL: https://creativecommons.org/licenses/by/4.0/

Keywords: Face matching, Metacognition, Training, Insight, Individual differences

Funding: The author received no funding for this work.

==============================
Background

Although researchers have begun to consider metacognitive insight during face matching, little is known about the underlying mechanism. Here, I investigated whether objective ability, as well as self-assessed ability, were able to predict metacognitive performance, that is, the ability to differentiate correct and incorrect responses in terms of confidence. In addition, I considered whether a training intervention resulted in improvements to both face matching performance and metacognitive insight.

Methods

In this experiment (N = 220), participants completed a face matching task, with either a diagnostic feature training course or a control course presented at the halfway point. In addition, a second face matching task, as well as a self-report questionnaire regarding ability, were completed to provide measures of objective and self-assessed ability respectively.

Results

Higher self-assessed ability with faces, as well as higher objective ability with face matching, predicted better metacognitive performance, i.e., greater confidence in correct, in comparison with incorrect, responses. This pattern of results was evident both when objective ability was measured through performance on the same task used to measure metacognitive insight and when a different task was used. Finally, the training intervention failed to produce improvements in face matching performance and showed no evidence of altering metacognitive ability.

Discussion

The current work begins to address the mechanism underlying individual differences in metacognitive insight during face matching. Although support was provided for a competence-based account, where better face matchers showed greater performance on the task and were more successful in monitoring their performance, further work might focus on decoupling task performance and competence in order to more conclusively explain why some people are more insightful than others.

Introduction

In a variety of situations involving the confirmation of identity, we are required to compare two images, or an individual and the document they present, and decide whether these show the same person or not. Known as ‘face matching’, this task is often difficult because the identities under consideration are unfamiliar to us (e.g., Bruce et al., 1999, 2001; Burton, White & McNeill, 2010; Kemp, Towell & Pike, 1997). As a direct result, we are forced to rely on the visual properties of the particular images (Hancock, Bruce & Burton, 2000) since we have no experience with additional information regarding the idiosyncratic variability of the face itself (Burton et al., 2016). Recent evidence has shown that super-recognisers (people who naturally have extraordinary face-processing skills) exhibit significantly higher accuracy, as well as greater confidence, in their face matching decisions than the average person (Bobak, Hancock & Bate, 2016), along with some self-awareness of their superior abilities (Bate & Dudfield, 2019). This raises the question, more broadly, as to whether people have differing levels of insight into their performance/competence (termed ‘metacognition’) on face perception tasks, and if so, why.

Metacognition and face processing

Previous research has found only moderate insight into one’s own face perception abilities, as assessed by the simple association between performance on face perception tasks and questionnaire-based self-estimates. For example, the 20-item prosopagnosia index (PI20; Shah et al., 2015a) has demonstrated medium-sized correlations in several studies (Gray, Bird & Cook, 2017; Livingston & Shah, 2018; Shah et al., 2015b; Ventura, Livingston & Shah, 2018; although for a possible qualification of this, see Estudillo & Wong, 2021) with performance on both the Glasgow Face Matching Test (GFMT; Burton, White & McNeill, 2010) and the Cambridge Face Memory Test (Duchaine & Nakayama, 2006). Other self-report measures have also shown similar medium-sized associations with additional matching and memory tasks (Bobak, Mileva & Hancock, 2019; Kramer, 2021; Matsuyoshi & Watanabe, 2021). Interestingly, when estimates are directly related to the task itself (e.g., “how will I perform on this matching test?”) rather than some general ability, the correlation remains only moderate (Zhou & Jenkins, 2020).

Evidence also suggests that people show insight into their performance at the level of individual trials, represented by higher confidence in their correct vs incorrect responses. Across participants, studies of face perception have found that the mean confidence rating for trials answered correctly was higher than for trials receiving incorrect responses (Bruce et al., 1999; Hopkins & Lyle, 2020; Stephens, Semmler & Sauer, 2017). At this group level of analysis, confidence appears to be able to differentiate between correct and incorrect responses for face processing tasks, including face recognition (Grabman & Dodson, 2020), searching for faces in crowds (Davis et al., 2018; Kramer, Hardy & Ritchie, 2020), and identifying faces that were present in previously shown arrays (Ji & Hayward, 2020).

Of course, not all participants are equally capable of differentiating between their correct and incorrect responses with regard to confidence. Across two different tasks, Kramer et al. (2022) found that increased competence with face matching (i.e., a higher overall score on objective tests) predicted greater metacognitive insight, represented by a larger distinction between correct and incorrect responses in terms of confidence ratings given. The best performers were significantly more confident in their correct (vs incorrect) responses, while the worst performers’ confidence ratings failed to distinguish between correct and incorrect decisions.

Although greater competence on a task predicted greater insight, the explanation for this remains unclear since the same task was used to both define objective ability (i.e., competence) and measure overall performance (see Grabman & Dodson, 2022). Because those with higher face matching ability also (by definition) performed better on the face matching task itself, the informativeness of confidence judgements may have been driven by either performance on the specific task or objective ability more generally. These two explanations have been referred to as the performance-based (or optimality; e.g., Deffenbacher, 1980) and competence-based (or decision processes; e.g., Kruger & Dunning, 1999) accounts respectively (for a review, see Grabman & Dodson, 2022). The competence-based account assumes that the same skills are necessary to both perform well on the task and successfully monitor one’s performance. In other words, as a result of being strong face matchers, such individuals are more accurate during the task and their confidence judgements are better informed (i.e., they show a larger separation between confidence in correct vs incorrect responses). In contrast, the performance-based account predicts that equating accuracy across strong and weak face matchers will produce similar levels of metacognitive insight across the range of objective ability. This is because performance on the task itself drives the informativeness of confidence judgements during the task. In order to distinguish between these two accounts, separate measures are needed for objective ability and task performance (e.g., through the inclusion of two different face matching tasks).

Grabman & Dodson (2022) also described a third potential explanation—the metacognitive awareness account—in which the association between objective and self-assessed abilities is key. The idea is that people who are attuned to their abilities (that is, their self-assessments are accurate) will show better metacognitive sensitivity (i.e., more informed confidence judgements). Therefore, if these two factors are aligned (e.g., those with poor objective ability also have low self-assessed ability) then metacognitive awareness is predicted to be high, resulting in greater confidence in correct vs incorrect decisions. Put simply, effectively monitoring one’s trial-by-trial performance can come from being either a weak face matcher who recognises their limited ability or a strong matcher who is well-aware of their ability in that domain. Evidence supporting this account would be provided by an interaction between objective and self-assessed abilities when considering metacognitive sensitivity. As such, a measure of self-assessed ability is needed—something which was absent in Kramer et al. (2022).

Face matching training and metacognition

Given that higher competence with face processing is potentially associated with greater metacognitive insight (Grabman & Dodson, 2022; Kramer et al., 2022), perhaps increases in competence result in improvements to metacognition. Simply put, could training participants to be better at face matching also improve their metacognitive monitoring (e.g., the ability to differentiate between their correct and incorrect decisions)?

Researchers have argued that, for a particular skill, competence and metacognitive insight rely on the same underlying process (Kruger & Dunning, 1999). As a result, those who perform poorly on a task also lack the metacognitive awareness to accurately assess their competence. Logically, according to this account, training participants to perform better should also produce improvements in their metacognitive insight. Kruger & Dunning (1999; Study 4) provided some preliminary evidence of this idea by showing that initial self-estimates of performance on a test of logical reasoning were subsequently improved after participants received a training intervention. However, it is worth noting that, although the intervention was designed to improve participants’ logical reasoning skills, this was never confirmed through follow-up testing.

While there are obvious reasons to investigate training interventions relating to face matching (e.g., to reduce passport officers’ errors at border control), only small progress has been made to date. Evidence suggests that professional training programs that are currently in use fail to produce improvements in performance (Towler et al., 2019). Experimentally, studies have typically attempted to increase accuracy by instructing participants to utilise a feature-by-feature comparison strategy (concentrating on individual features, rather than the face ‘as a whole’), although this approach has garnered mixed results (Megreya, 2018; Megreya & Bindemann, 2018). The reason for this may be the lack of clarity regarding which features participants should prioritise. For example, Megreya & Bindemann (2018) found that eyebrow comparisons may be useful, although this could depend on the particular photosets under consideration.

Further investigating this idea of instructing participants to focus on particular features when face matching, Towler et al. (2021) implemented an evidence-based training intervention. Participants were instructed, through a series of slides, to prioritise the ears and facial marks since these had previously been shown to be most useful according to facial examiners, whose job it is to carry out these kinds of facial comparisons (Towler, White & Kemp, 2017). Across two studies, the training resulted in an accuracy increase of around 6% post-intervention. Although a relatively small increase, this represents approximately half of the difference in performance previously found when comparing novice participants with facial examiners (Towler, White & Kemp, 2017). Indeed, this training method has also been shown to improve accuracy when matching full faces to ones wearing face masks (Carragher et al., 2022). Therefore, this training approach appears to be the most robust to date.

The current experiment

Considering the unanswered questions highlighted above, the current experiment aimed to make some progress with regard to face matching and metacognition. First, extending the work of Kramer et al. (2022), I aimed to better understand the mechanisms underlying the relationship between trial-level confidence and face matching accuracy by including an objective measure of ability separate from the task under consideration, as well as a measure of self-assessed ability. These two additions allowed for a comparison of the different metacognitive accounts which was lacking in the original study. Second, by replicating the design of Towler et al. (2021), I aimed to produce improvements in face matching accuracy through a training intervention. As a result, I would be able to investigate whether this intervention, and predicted performance increase, might also produce an increase in metacognitive insight.

Method

Participants

After restricting eligibility to those located in countries where the majority of residents speak English (i.e., the USA, the UK, Canada, New Zealand, and Australia), 344 participants were recruited through Amazon Mechanical Turk (MTurk). Of these, 220 (103 women; age M = 37.5 years, SD = 10.8 years; 90.5% self-reported ethnicity as White) completed the study and correctly answered all six of the attention checks. Participants were paid US $3.50, which approximated a rate of $7 per hour. Each participant provided written, informed consent online before taking part, and received an onscreen debriefing upon completion, in line with the ethics protocol of the university. Ethical approval was granted by the University of Lincoln’s ethics committee (ID 9130).

I conducted an a priori power analysis using G*Power 3.1 (Faul et al., 2007). In order to detect the effect of diagnostic feature training on performance, I used the raw data provided by Towler et al. (2021; Experiment 1). After recalculating the training (diagnostic feature, control) × test (pre-training, post-training) ANOVA as adapted to the current experiment (i.e., excluding the nondiagnostic feature training condition), the resulting effect size of the interaction (ηp2 = 0.19) and the correlation between the repeated measures (r = 0.58) were used to determine sample size. In order to achieve 95% power at an alpha of 0.05, a total sample size of 16 was required.

Materials

A recently developed version of the GFMT (GFMT2-SA; White et al., 2022) was used to measure objective face matching ability. The task comprised 40 pairs of faces, where half of the pairs were match trials (the same person in different images) and half were mismatch trials (different but similar-looking people). All images were presented in colour, and incorporated changes in expression, head angle, and camera-to-subject distance. This version of the test was chosen in order to provide a face matching task of comparable difficulty (74.5% accuracy in previous work; White et al., 2022) to the second test chosen.

This second face matching test, the Expertise in Facial Comparison Test (EFCT; White et al., 2015) consisted of 168 pairs of faces (84 match and 84 mismatch trials), where subjects in the images were photographed multiple times on different days in unconstrained, naturalistic settings. Average item accuracy in a previous study was 75.4% (White et al., 2015). Following Towler et al. (2021), the EFCT was divided into two sets of 84 trials (42 match and 42 mismatch trials) known to be of equal difficulty (labelled A and B).

Along with these two EFCT subtests, participants completed one of two training courses used by Towler et al. (2021). The diagnostic feature training course comprised 15 slides that instructed participants to “avoid viewing the face as a whole”, “compare each feature individually”, and focus on the ears and facial marks (with additional details also provided). This content was derived from previous research (Towler, White & Kemp, 2017) and has been shown to improve face matching performance (Towler et al., 2021). To serve as a control condition, Towler’s conflict resolution training course, also comprising 15 slides, was used. This course provided no information that could aid face matching performance.

I used the PI20 (Shah et al., 2015a) to measure participants’ self-assessed abilities with face recognition. (To date, there is no questionnaire designed to measure self-assessed face matching ability). For each item, participants selected a response from the following: strongly agree, agree, neither agree nor disagree, disagree, strongly disagree. After reverse coding five items, overall score was calculated by summing individual responses (possible range: 20–100), with lower scores indicating higher self-reported estimates of face recognition ability. Scores on this test averaged 42.0 in the general population (Shah et al., 2015b). The PI20 has demonstrated high levels of internal consistency (Cronbach’s α = 0.93; Shah et al., 2015b) and was chosen because scores were strongly correlated with face matching ability (r = −0.49; Shah et al., 2015b).

Procedure

The experiment was carried out using the Gorilla online testing platform (Anwyl-Irvine et al., 2020). Information was collected regarding the participant’s age, gender, and ethnicity, as well as their MTurk Worker ID. This Worker ID allowed ‘qualifications’ to be assigned, which were used to prevent participants from taking part on multiple occasions. Participants were also prevented from using mobile phones or tablets (via settings available in Gorilla) to ensure that images were viewed at an acceptable size onscreen.

Participants first completed the PI20 questionnaire to ensure that their experience with the face matching tests did not affect their self-estimates of ability. Next, they completed the GFMT2-SA. On each trial, participants decided whether the two images depicted the same person or two different people (using a two-alternative forced choice). They were then asked how confident they were in their response, using a 0 (not at all confident) to 5 (extremely confident) scale. In addition to the original 40 trials, two attention checks were inserted within the test. These were included because attentiveness is a common concern when collecting data online (Hauser & Schwarz, 2016). One of these attention checks featured two identical images of a celebrity (e.g., Ryan Reynolds) in order to provide an easy match trial. The other attention check featured one image of each of two celebrities who differed in both gender and ethnicity (e.g., Will Smith and Scarlett Johansson), providing an easy mismatch trial. The presentation order of the 42 trials was randomised for each participant.

Next, and following the general procedure of Towler et al. (2021), participants completed one subtest of the EFCT (A/B), then a training course (either diagnostic feature training or the control course), and then the other EFCT subtest (B/A). In both subtests, participants responded as in the previous test (i.e., a same/different forced choice response followed by a confidence rating). The order of subtests was randomly allocated for each participant, as was the specific training course. In each of the two subtests, two attention check trials were also inserted (see above) and trial order was randomised for each participant.

Responses during the face matching tests were self-paced, without time limits, and no feedback was given. Progression through the training course was also self-paced, with participants expected to take around 5.5 min to read through the materials (Towler et al., 2021).

Finally, to aid transparency, I note that all measures and conditions have been reported here.

Results

Data analysis included only those participants who correctly answered all six of the attention checks (see above for details). In this final sample, 107 participants completed the control training (48 women; age M = 38.0 years, SD = 10.1 years; 91.6% self-reported ethnicity as White), while 113 participants completed the diagnostic feature training (55 women; age M = 37.0 years, SD = 11.5 years; 89.4% self-reported ethnicity as White). These participants took approximately half an hour (M = 27.3 min, SD = 9.1 min) to complete the whole experiment from start to finish.

In terms of overall performance on the tasks, scores on the GFMT2-SA (percentage correct M = 75.0%, SD = 10.5%; sensitivity d′ M = 1.57, SD = 0.67) were very similar to those found in previous work (74.5%; White et al., 2022). Performance on the EFCT, considering the whole test and irrespective of training condition (percentage correct M = 67.1%, SD = 9.1%; sensitivity d′ M = 1.09, SD = 0.53), was a little lower than in previous work (75.4% based on average item accuracy; White et al., 2015). Finally, scores on the PI20 (M = 53.8, SD = 15.1) were a little higher than previously reported in the general population (42.0; Shah et al., 2015b), suggesting poorer self-reported face recognition abilities in this sample of participants.

Does training influence performance and metacognition?

I sought to first replicate the findings of Towler et al. (2021) by investigating whether participants had shown improvement on the EFCT as a result of completing the diagnostic feature training. For each subtest of the EFCT, in line with previous work in this field (e.g., Kramer & Ritchie, 2016), I calculated percentage correct scores, as well as a measure of sensitivity (d′).

Following Towler et al. (2021), I analysed the data using 2 (training: diagnostic feature, control) × 2 (test: pre-training, post-training) mixed ANOVAs, where training varied between participants and test varied within participants.

For percentage correct scores, I found a main effect of test, F(1, 218) = 18.01, p < 0.001, ηp2 = 0.08, with participants scoring higher pre-training (M = 68.2%, SD = 0.7%) in comparison with post-training (M = 66.0%, SD = 0.7%). However, there was no main effect of training, F(1, 218) = 0.14, p = 0.710, ηp2 < 0.01, and no interaction between these two factors, F(1, 218) = 1.58, p = 0.210, ηp2 = 0.01.

For sensitivity d′, I found a main effect of test, F(1, 206) = 13.07, p < 0.001, ηp2 = 0.06, and no main effect of training, F(1, 206) = 0.01, p = 0.933, ηp2 < 0.01. However, these results were qualified by a significant interaction between the two factors, F(1, 206) = 4.66, p = 0.032, ηp2 = 0.02. I therefore considered the simple main effects of test at each level of training. For the control training, participants performed significantly worse post-training (M = 1.03, SD = 0.06) in comparison with pre-training (M = 1.23, SD = 0.06), F(1, 98) = 16.60, p < 0.001, ηp2 = 0.15, while those who completed the diagnostic feature training (M = 1.12, SD = 0.05) did not differ across tests, F(1, 108) = 1.08, p = 0.302, ηp2 = 0.01.

Despite the lack of evidence that diagnostic feature training improved performance, using either percentage correct or d′, I considered whether this training intervention may still have influenced metacognition. To this end, I investigated individual trial responses (the same/different decision and its associated confidence rating) as a measure of metacognitive insight—a larger difference in confidence between correct and incorrect responses would reflect greater insight. I used linear mixed-effects models, following on from previous work (Grabman & Dodson, 2022; Kramer et al., 2022), and statistical analyses were carried out using R (lme4 package—Bates et al., 2015). For significance reports, degrees of freedom were estimated using Satterthwaite’s method (lmerTest package—Kuznetsova, Brockhoff & Christensen, 2017).

As described in Kramer et al. (2022), I used crossed random effects (because each participant completed the same series of trials), where participants and trials variance were considered at Level 2 and residual variance at Level 1. In terms of the data set, each participant by trial observation was the unit of analysis, with each row of data indicating the training (diagnostic feature, control) and test (pre-training, post-training) for the participant’s particular trial, the confidence rating given by the participant to that trial (trial confidence), and whether the response given was correct or incorrect (trial accuracy). The model predicted the confidence value assigned to each trial from the fixed effects of training, test, trial accuracy, and all interactions between these factors. In this model, only the intercept varied randomly across trials, whereas the intercept and the slope of trial accuracy varied randomly across participants. Models using more complex random effects structures were identified as singular (Barr et al., 2013).

I found a significant main effect of trial accuracy, ß = 0.26, SE = 0.03, t(305) = 7.81, p < 0.001, where higher confidence ratings were given to correct, in comparison with incorrect, responses. The main effects of training, ß = 0.20, SE = 0.11, t(228) = 1.90, p = 0.059, and test, ß < 0.01, SE = 0.02, t(36620) = 0.13, p = 0.898, were not statistically significant. (Note that if the effect of training had been significant, it would simply have shown that higher ratings of confidence were given by participants who had received the diagnostic feature training). Most importantly, none of the interactions were significant (all ps > 0.185), suggesting that the relationship between a trial’s confidence and accuracy was not influenced by the type of training that participants received.

Self-assessed ability, objective ability, and metacognitive sensitivity

Next, using a similar trial-level approach, I investigated whether objective and self-assessed abilities predicted which participants showed better insight.

First, I considered participants’ objective ability and metacognitive sensitivity during the same task—the GFMT2-SA. The model predicted the confidence value assigned to each trial (trial confidence) from the fixed effects of self-assessed ability (PI20 score, standardised), objective ability (GFMT2-SA score, standardised), trial accuracy (correct vs incorrect), and all interactions between these factors. In this model, only the intercept varied randomly across trials, whereas the intercept and the slope of trial accuracy varied randomly across participants. Models using more complex random effects structures were identified as singular (Barr et al., 2013).

I found significant main effects of trial accuracy, ß = 0.36, SE = 0.03, t(201) = 10.87, p < 0.001, and self-assessed ability, ß = 0.01, SE < 0.01, t(210) = 2.69, p = 0.008. However, these effects were qualified by significant two-way interactions: trial accuracy × objective ability, ß = 1.14, SE = 0.33, t(188) = 3.47, p < 0.001; trial accuracy × self-assessed ability, ß = −0.01, SE < 0.01, t(187) = −4.77, p < 0.001. As Fig. 1A illustrates, for those with higher objective ability, confidence ratings better discriminated between correct and incorrect responses. The same pattern was also shown by those with higher self-assessed ability (i.e., lower PI20 scores; see Fig. 1B). The remaining two- and three-way interactions were not statistically significant (both ps > 0.777), suggesting separate effects of objective and self-assessed abilities on metacognitive sensitivity.

Figure 1 An illustration of GFMT2-SA confidence as a function of (A) objective ability on the same task and (B) self-assessed ability.

Separate lines represent correct and incorrect responses. Error bands represent 95% confidence intervals. Lower scores on the PI20 indicate higher self-reported estimates of face recognition ability.

Next, I considered participants’ objective ability on one task (GFMT2-SA) as a predictor of metacognitive sensitivity during a different task (EFCT), allowing for the distinction between objective ability and current task performance. Following the same analytical approach as before, I included only participants’ responses during the first subtest of the EFCT that they completed (pretraining), irrespective of which subtest it was. The fixed effects were self-assessed ability (PI20 scores, standardised), objective ability (GFMT2-SA scores, standardised), trial accuracy (correct vs incorrect), and all interactions between these factors. Again, only the intercept varied randomly across trials, whereas the intercept and the slope of trial accuracy varied randomly across participants. Models using more complex random effects structures were identified as singular (Barr et al., 2013).

I found a significant main effect of trial accuracy, ß = 0.23, SE = 0.02, t(197) = 9.43, p < 0.001. However, this was qualified by significant two-way interactions: trial accuracy × objective ability, ß = 0.54, SE = 0.23, t(191) = 2.33, p = 0.021; trial accuracy × self-assessed ability, ß = −0.01, SE < 0.01, t(188) = −4.26, p < 0.001. Figures 2A and 2B show the same patterns as those for the previous analysis—for participants with higher objective ability, and separately for those with higher self-assessed ability, confidence ratings better discriminated between correct and incorrect responses. As above, the remaining two- and three-way interactions were not statistically significant (both ps > 0.299).

Figure 2 An illustration of EFCT confidence as a function of (A) objective ability on the GFMT2-SA and (B) self-assessed ability.

Separate lines represent correct and incorrect responses. Error bands represent 95% confidence intervals. Lower scores on the PI20 indicate higher self-reported estimates of face recognition ability.

Discussion

This experiment adds to the sparse literature on face matching and metacognition. To date, research has begun to establish that individual differences exist in metacognitive insight, and that these are related to ability (Kramer et al., 2022), but little is known about the underlying mechanism. Here, I provide some further support for the competence-based account, arguing that people who have greater objective ability on a task should demonstrate greater metacognitive insight (in line with previous work on face recognition; Grabman & Dodson, 2022).

I found that participants with higher objective ability, and separately those with higher self-assessed ability, better discriminated between correct and incorrect responses in terms of their confidence ratings. This pattern was apparent when objective ability was defined using either competence on the task under consideration (i.e., the GFMT2-SA) or competence on a different task (GFMT2-SA competence when considering trial accuracy and confidence on the EFCT). This makes intuitive sense given that performance on the two tasks were strongly correlated (r = 0.61 for percentage correct; r = 0.58 for d′ sensitivity). In order to differentiate between the performance-based and competence-based accounts, accuracy on the task must be separated from objective ability (see Grabman & Dodson, 2022). Of course, more competent performers typically produce higher task accuracies, meaning that it could be either of these two factors that explains greater metacognitive insight on the task. Here, by using a measure of objective ability taken from a separate task, my results were more supportive of the competence-based account—that is, greater competence predicted better metacognitive insight (in line with recent work in face recognition; Gettleman et al., 2021; Grabman & Dodson, 2022).

The results also provide little support for the metacognitive awareness account of individual differences in the confidence–accuracy relationship. This account describes the possible association between objective and self-assessed abilities. If these two factors are aligned then metacognitive awareness is predicted to be high, i.e., for those who show low competence and also assess themselves as low ability, or for those who are highly competent and report high self-assessed ability. Evidence of this account would require an interaction between objective and self-assessed abilities (and trial accuracy). However, this was absent in the current experiment, providing no evidence in support of this particular account and mirroring previous work in face recognition (Grabman & Dodson, 2022). Indeed, the current findings suggest separate influences of objective and self-assessed abilities on metacognitive performance. Since the two measures only showed a moderate association in this study (e.g., r = −0.32 between GFMT2-SA percentage correct and PI20 score), the implication is that individuals could demonstrate lower metacognitive awareness alongside higher objective ability, for instance.

Interestingly, if we compare participants across objective (or self-assessed) abilities, we see that confidence on correct responses remained relatively unchanged (see Figs. 1 and 2). However, confidence in their incorrect responses decreased for better face matchers (or those who assessed themselves as better). This pattern differs from that of Kramer et al. (2022), where incorrect response confidence was similar across all participants, while confidence on correct responses increased for better performers. Therefore, unlike in previous work, the data here suggest that weak performers (and those who assessed themselves as worse) were overconfident in their incorrect responses, which arguably aligns with the findings described by the Dunning–Kruger Effect. In their work, Kruger & Dunning (1999) showed that the weakest performers overestimated their overall test performance, which implies misplaced confidence. As such, although several studies have identified serious flaws with the general approach utilised by Kruger and Dunning (e.g., Gignac & Zajenkowski, 2020; Krajc & Ortmann, 2008; Krueger & Mueller, 2002; Magnus & Peresetsky, 2022; Meeran, Goodwin & Yalabik, 2016; Nuhfer et al., 2016, 2017), the notion that weaker performers are overconfident in their performance may be a reliable one.

It is worth noting that the current set of results come with a caveat. Given the large performance correlations between tasks, I acknowledge that competence and performance have not been entirely decoupled. Ideally, a task is needed that results in a range of accuracies from both good and poor face matchers. For face recognition, this has been achieved through purposely manipulating the accuracy of line-up identifications by varying encoding exposure durations and retention intervals (Gettleman et al., 2021; Grabman & Dodson, 2022). Although these variables are not applicable to a face matching paradigm, one could manipulate task difficulty through, for example, decreasing image quality (Bindemann et al., 2013) or presentation times (Bindemann, Avetisyan & Blackwell, 2010). This represents a promising route for future study.

Disappointingly, the training intervention failed to improve face matching performance, although there was some evidence to suggest that the apparent drop in d′ sensitivities from pre- to post-training was prevented because of the diagnostic feature training. Further, and perhaps due to this lack of improvement in performance, the intervention had no effect on metacognitive sensitivity. As a result, I was unable to investigate whether increased competence would produce increased insight. I can only speculate on the reasons for the failure to replicate previous work (Towler et al., 2021). The most salient differences between this experiment and the original study include, first of all, the sample size. The participant ratio for control training to diagnostic feature training in the original study was 20:20, in comparison with 107:113 here. Second, the original study did not include any attention checks to confirm that participants were paying attention throughout the task, although this may have been unnecessary since their task was completed in person rather than online. However, it is worth noting that the same training intervention was successful in improving performance with a task involving face masks when run with paid, online participants and including attention checks (Carragher et al., 2022). Finally, Towler et al. (2021) measured both pre- and post-training performance using area under the curve (AUC), while percentage correct and d′ sensitivity were considered here. The current experiment better suited separate measures of trial accuracy and confidence in order to investigate the relationship between the two, while Towler et al. (2021) utilised a five-point response scale (1 (sure same person) to 5 (sure different people)), which allowed for the calculation of AUC. However, as above, at least with a task involving face masks, the same training intervention led to improvements in both percentage correct and d′ sensitivity (although these were calculated after collapsing across their six-item response scale; Carragher et al., 2022).

In sum, for the task of face matching, where both accuracy and confidence are crucial factors to consider, and decisions can have far-reaching security implications, individual differences in metacognition represent an important avenue that has so far received little attention. I found support for the competence-based account, which ascribes differences in metacognitive insight in face matching tasks to underlying differences in objective ability. While the reason for the lack of improvement with the training intervention remains unclear, the solution may be to develop more robust interventions that produce larger improvements in performances. If results hold, this increment in performance will carry the secondary benefit of improving metacognitive insight.

Supplemental Information

Supplemental Information 1 Raw data for all analyses.

The data show that the training manipulation failed to improve performance or affect metacognitive ability, and that higher self-assessed and objective abilities predicted better metacognitive performance.

Click here for additional data file.

The author thanks Jesse Grabman for his critical comments on the manuscript.

Additional Information and Declarations

Competing Interests

Author Contributions

Human Ethics

Data Availability

Robin S. S. Kramer is an Academic Editor for PeerJ.

Robin S. S. Kramer conceived and designed the experiments, performed the experiments, analyzed the data, prepared figures and/or tables, authored or reviewed drafts of the article, and approved the final draft.

The following information was supplied relating to ethical approvals (i.e., approving body and any reference numbers):

Ethical approval for this experiment was granted by the University of Lincoln’s ethics committee (ID 9130).

The following information was supplied regarding data availability:

The raw data are available in the Supplemental File.

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
