# Peer review of "Face matching and metacognition: investigating individual differences and a training intervention"

_PeerJ, doi:10.7717/peerj.14821_

## Round 0.1 · original submission · Minor Revisions

First, let me apologize for the extreme delay in a decision on your manuscript. This is a difficult time of year in which to find reviewers. Despite the challenge, I have received reviews from two experts in the field. I thank them for their willingness to offer their expertise and for the time they put into critiquing the manuscript!

You will see that both reviewers are quite positive about the manuscript, as am I. I believe their concerns should be rather easily addressed in a revision. I would like to highlight one theme that emerged from the reviews: Both reviewers noted that they saw value in comparing metacognition across training conditions. Even if the training doesn't improve performance, it could still impact metacognition. I, too, would be curious to know the outcomes of such an analysis, and I think it would make a valuable addition to the paper.

Finally, I request that you add a statement to the paper confirming whether you have reported all measures, conditions, data exclusions, and how you determined your sample sizes. You should, of course, add any additional text to ensure the statement is accurate. This is the standard reviewer disclosure request endorsed by the Center for Open Science [see http://osf.io/project/hadz3]. I include it in every review.

Best,
Tony Barnhart

Reviewer 1 ·

Basic reporting

The manuscript is clearly written. In my view, it would benefit from including some extra literature that has explored this topic before (see my comments in 4)

Experimental design

Very relevant questions which is investigated in a rigorous manner. In my view, the description of the tasks should include some extra details (e.g., how participants provided responses to the matching tasks, etc).

Validity of the findings

The rationale of the study is clear. Raw data are provided. Conclusions are supported by the analysis.

Additional comments

Dear Editor,
Thanks a lot for the opportunity to review this paper. In the manuscript “Face matching and metacognition: Investigating individual differences and a training intervention” the author presents a single experiment exploring the relationship between individual differences in face matching abilities and metacognitive insights to solve this task. The results showed that higher objective face matching accuracy and self-reported ability was associated with higher confidence during matching performance. In my view, the paper is nicely written and could make a nice contribution to the journal. I have a couple of comments.
In my view the intro, needs a bit more of context. For example, the relationship between different levels of self-reported identification skills and face identification performance has been previously studied. Bate and Dudfield (2019) found that super-recognizers have better insights into their face identification skills. Estudillo and Wong (2021) that the relationship between the PI20 and objective face identification is driven mainly by good and bad recognizers.
I also have a comment regarding the use of the PI20. I understand that this questionnaire has been used previuosly to explore individual differences in self-reported face identification. However, the questionnaire was not created for this but to help the diagnosis of prosopagnosia (see Tsantani et al., 2021). In fact, when developmental prosopagnosics (DP) are tested the PI20 explains around 46% of the variance of the Cambridge Face Memory Test, but when the sample does not include DPs, this variance decreases to between 5-15% (e.g., Matsuyoshi & Watanabe, 2020). Taking this -and what I mentioned in the previous paragraph- into account, using the PI20 could be a limitation of the current study that in my view should be discussed.
In contrast to previous research, the training program did not work. Wouldn’t be still interesting to explore whether participants in the training group show a change in their metacognitive insight after the training?

Reviewer 2 ·

Basic reporting

The article is clear (with some suggestions for improved clarity below), and professional English is used throughout. The literature cited provides sufficient overview of the topic. The article has a conventional structure, and professional figures. The raw data has been shared. The research questions clearly follow the rationale of the introduction.

The abstract is reasonably clear. Some minor revisions might help to improve clarity.
1) Line 17 – “metacognitive performance” is not clearly explained/defined.
2) Perhaps lines 28 and 29 might be revised to clarify that metacognitive improvements were not considered in the context of the (unsuccessful) training intervention.
3) Line 31 – the “competence-based account” has not been introduced yet in the abstract.
4) Perhaps related to the previous point, the abstract mentions that the study is about testing the mechanisms underlying individual differences in metacognitive insight, but this point is not elaborated on in more detail (i.e., what are the potential mechanisms? Why do these findings support the competence-based account?). Perhaps further detail could be added to the “discussion” in the abstract?

Likewise, the introduction is also generally well written, and gives a good overview of the relevant literature. Again, I only have some minor points for the author to consider revising.
1) The current experiment seems reasonably similar to the author’s previous work (Kramer et al., 2022) – but the difference between the two papers could be clearer.
2) The explanations offered for the “competence-based” and “performance-based” accounts of metacognition could be expanded on – I find the performance based account a little difficult to follow (lines 88-89). Perhaps some examples related to face matching might help to illustrate the differences between the two? (Lines 90-91 of the same paragraph are much clearer because of the example). Likewise, I’m not sure I follow the logic presented in Lines 95-96 since throughout the rest of the manuscript, the point is made that better cognitive awareness is given by greater separation in confidence between correct and incorrect responses – is the assumption in the example that someone who knows they are not very good at the task will still give higher confidence ratings when they are certain they are correct? How does that fit with the notion that they are not good at the task (i.e., by definition making lots of incorrect judgments)?

Experimental design

The experimental design is clear and appropriate. I only have some minor points that I'd like to see addressed in the text.

a. What were the participant demographics in each training condition?
b. Perhaps just check the specifications of the power analysis – using G*Power, my analysis using the recommended effect size specification (“options” > “Cohen’s”), and the values stated in the paper, returns a total of 58 participants. It’s a moot point since you have collected a much larger sample, but I’d just recommend checking the power analysis.
c. How long did the experiment take to complete (on average)?
d. One thing noted in the general discussion is that both prior papers showing the effectiveness of diagnostic feature training (Towler et al., 2021; Carragher et al., 2022) had participants make a single response to each trial, which conveyed their identification decision (same/different) and confidence. Is there a reason that the two judgments were made separately in the current study? It there any possibility that this difference might contribute to the failed replication of the training intervention?

Validity of the findings

As above, the results are clearly reported and well written. I only have some minor points that I'd like to see clarified.

a) Results – I’d like to see the basic performance data reported for the GFMT2-SA and EFCT. Did the online participants produce performance similar to that expected based on the norms for these tests? From the means given in Lines 245-246, these means might be a little lower than expected.
b) Results – Could you please add standard deviation to the means that are given for the main effects?
c) Results – The trend seems to be for performance to decrease from pre-training to post-training for the EFCT. First, is there any explanation for this finding? Second, could something perhaps be said for the diagnostic training potentially preventing the significant decrease in performance that was observed in the control condition? (And could you please give the M/SD for the diagnostic training condition – Line 254?). Finally, to follow on from my point above – while the diagnostic feature training condition did not improve, the interaction between Training and Test was still significant, suggesting that the training has had some kind of effect. Is it necessary for the training condition to improve before investigating whether the training has had some kind of effect on metacognition? Why wouldn't it be valid to investigate whether training had an effect on metacognition, particularly when the interaction term is significant? This is only a suggestion – even if not implemented by the author, I think the justification could be stronger in the text.
d) Could the y-axis on the figures be standardised? And perhaps a reminder could be added to the figure caption that lower scores indicate better performance on the PI20?

Additional comments

The author makes an interesting point on Line 288 – that objective and self-assessed abilities have separate effects on metacognitive sensitivity. So that suggests there are individuals with high objective performance, but who have poor metacognitive awareness? Can this point be expanded on in the general discussion?

---

## Round 0.2 · accepted · Accept

Thanks for resubmitting this work. Given that all of the original reviews were quite positive (and the requested revisions were minimal), I have not sent the manuscript out for re-review. After a thorough re-reading of your work, I am satisfied that you have addressed all of the reviewers' concerns and that the piece meets threshold for publication at PeerJ.

Congratulations!

Tony Barnhart